# The Impact of Biomedical Engineering on the Development of Minimally Invasive Cardio-Thoracic Surgery

**DOI:** 10.3390/jcm10173877

**Published:** 2021-08-28

**Authors:** Riccardo Cocchieri, Bertus van de Wetering, Marco Stijnen, Robert Riezebos, Bastian de Mol

**Affiliations:** 1Heart Center, OLVG Hospital, 1091 AC Amsterdam, The Netherlands; R.Cocchieri@olvg.nl (R.C.); R.K.Riezebos@olvg.nl (R.R.); 2Department of Biomedical Engineering, Eindhoven University of Technology, 5600 MB Eindhoven, The Netherlands; m.stijnen@lifetecgroup.com; 3LifeTec Group BV, 5611 ZS Eindhoven, The Netherlands; 4Department of Cardiothoracic Surgery, Amsterdam University Medical Center, 1105 AZ Amsterdam, The Netherlands

**Keywords:** minimal injury cardiac surgery, biomedical engineering, adaption and redesign, innovation of tools, R&D trade-offs

## Abstract

(1) We describe the boundary conditions for minimally invasive cardiac surgery (MICS) with the aim to reduce procedure-related patient injury and discomfort. (2) The analysis of the MICS work process and its demand for improved tools and devices is followed by a description of the relevant sub-specialties of bio-medical engineering: electronics, biomechanics, and materials sciences. (3) Innovations can represent a desired adaptation of an existing work process or a radical redesign of procedure and devices such as in transcutaneous procedures. Focused interaction between engineers, industry, and surgeons is always mandatory (i.e., a therapeutic alliance for addressing ‘unmet patient or professional needs’. (4) Novel techniques in MICS lean heavily on usability and safe and effective use in dedicated hands. Therefore, the use of training and simulation models should enable skills selection, a safe learning curve, and maintenance of proficiency. (5) The critical technical steps and cost–benefit trade-offs during the journey from invention to application will be explained. Business considerations such as time-to-market and returns on investment do shape the cost–benefit room for commercial use of technology. Proof of clinical safety and effectiveness by physicians remains important, but establishing the technical reliability of MICS tools and warranting appropriate surgical skills come first.

## 1. Introduction: The Inevitable Interaction between Biomedical Engineering and Cardiac Surgery

In their paper in the American Journal of Surgery, July 1967 entitled ‘Biomedical Applications to Cardiovascular Surgery’, Dagget and Austen extensively described the dependence of progress in cardiovascular surgery on biomedical engineering [1]. The paper presents a very readable and complete overview of how electronics, synthetics, mechanics, hydraulics, and metallurgy provided a solid technical basis for cardiac surgical equipment and devices. The invention of the cardiopulmonary bypass machine and the improvement in membrane technology have enabled safe and controlled cardiovascular interventions. Although very skilled, cardiac surgeons needed the support of extracorporeal circulation (ECC), in order to learn and to improve cardiac surgery, which in turn resulted in the continuing development of technology and devices. Modern technology and current experience and skills have reached such a high level that small access- and off-pump interventions are possible at low risk, and with equally good or better results than their more conventional counterpart. Cardiac surgical departments host their own R&D institutes in order to bring together innovations in engineering and affiliated disciplines, not only for practical needs, but also for the purposes of teaching and science [2]. Since 2006, the Methodist Hospital cardiovascular group based in Houston, Texas has hosted an annual ‘Pumps and Pipes’ meeting. Engineers from many fields including the aerospace, oil, and automotive industries present their views on innovation in health care technology (www.pumpsandpipes.org, accessed on 13 June 2021). The current influence of biomedical engineering on surgical platforms is overwhelmingly similar to the impact of engineering on other sections of society. Therefore, attempts to analyze the impact of a single discipline or technology in detail are futile and selective. Progress in minimally-invasive cardiac surgery (MICS) leans heavily on the newest instrumentation, devices, and imaging technology. This development forces surgeons to continuously reflect on their skills and surgical techniques, and their role in the safe implementation of biomedically-engineered technology for the patient’s benefit. 

### 1.1. Boundary Conditions of Minimally-Invasive Cardiac Surgery

Prior to clarifying what biomedical engineering can do for the evolution of MICS, it is necessary to describe what in engineering terms are called the ‘boundary conditions’ of design, process control, and usability.

Why is minimally-invasive cardiac surgery considered the exception rather than the rule in contrast with many areas of general surgery? The large-scale implementation of cardiac surgery and its potentially lethal associated risks require well-controlled standardized procedures with predictable outcomes and a satisfactory potential for life-saving bail-outs. Median sternotomy and ECC support provide these conditions. In addition, many of today’s indications for cardiac surgery are based on non-eligibility for a ‘simple’ percutaneous solution, or the need for concomitant intervention. Last but not least, the care of the sequelae of median sternotomy and ECC is very much improved in both low-complex and high-complex cases. 

### 1.2. Benefits of Minimally-Invasive Cardiac Surgery

Besides a faster recovery, what is there to gain from MICS (which makes the surgery more difficult) when the results of more conventional surgical techniques are already very good? It appears hard to respond to this question with appropriate evidence. Head-to-head comparison of surgical techniques by means of a randomized controlled trial (RCT) has rarely been undertaken, and comparing surgical beliefs is a proven recipe for unproductive debate. Therefore, the vast majority of cardiac surgeons are trained for, and will continue to be trained for, the ‘major-invasive approach’. Skills and competence in MICS and surgery-by-wire are considered to be additional areas of expertise. 

Supported by progress in imaging techniques and the expanding body of surgical experience, cardiac surgeons continue to attempt more difficult access routes to the heart and thoracic structures in order to reduce injury to structures and blood loss. Techniques originating from laparoscopy including smaller incisions and techniques to avoid a median sternotomy have been adopted. See Figure 1 for a typical overview of a MICS setting. These include video-display and adjusted light and lens performance based on glass fiber transfer and are enabled by porthole and mini-incision technology, specifically in mitral valve surgery [3].

In cardiac surgery, exposure to extracorporeal circulation and aortic cross-clamping is regarded as potentially injurious, and reduction in or avoidance of these procedures is also considered less invasive [4]. Today, MICS is established and safe practice in the dedicated hands of a minority of cardiac surgeons, whilst ‘micro-invasive’ cardiac surgery is the next phase where surgeons meet transcutaneous-driven techniques practiced by cardiologists.

### 1.3. MICS Progress: From Technique to Engineering-Based Technology

The results of biomedical engineering are now so completely embedded into cardiac surgical practice that separating them is no longer possible. This review aims to show that the progress and development of MICS will largely depend on engineering principles including human factor control, and not by trial and error and the conventional surgical virtues such as the development of skills, resourcefulness, serendipity, and risk-taking. To quote d’Onofrino: 

“… ‘Technology’ not ‘technique’ is evolving; let us adapt to this evolution keeping the ‘technique’ securely in our hands.” [5]. 

This means that surgeons have to put technology first, but the development of appropriate skills must follow. If used inappropriately, all promising technology is doomed to fall short. 

Apart from MICS, the cardiac surgeon’s toolbox is also expanding to include wire-mounted interventions such as trans-apical access, which enables aortic and mitral interventions as well as trans-aortic access for aortic valve replacement by the implantation of balloon- or self-expanding valves. 

### 1.4. Human Factor Engineering

Access by means of puncture or a significant incision determines whether a procedure is in the hands of the cardiologist or the cardiac surgeon. Although many cardiovascular surgeons combine wire procedures with surgery, cardiologists limit themselves to a small incision, creating a pacemaker pocket, for instance. Surgeons tend to prefer the rather time-consuming challenges of access and complexity. From the engineering point of view, the choice is simple: the operator with the best skills in handling the technology and the best potential to generate patient benefit should be in charge. This opinion is based on the principle that design requirements such as usability, user-friendliness, and predictability should come first. In reality, this technical human factor concept is a question of which specialist represents the highest business value in the professional arenas of hospitals and industry. Cardiology represents a larger market and far more value, thanks to the supporting and surrounding technology such as imaging, arrhythmia treatment, and disease management. We must be aware that amongst many other factors, business and market considerations determine whether a technology reaches the implementation stage and can become a medical success.

## 2. Biomedical Engineering Disciplines 

In this section, we explain which disciplines are the most relevant to conducting safe and effective interventions in accordance with the perioperative timeline. *Bioengineering* describes interactions with cells including blood, while biomedical engineering is mostly used to describe the interaction with structures [6]. Distinguishing between these disciplines is not always straightforward, and depends on the application. The novel coating of an introducer sheath or coronary stent should resist friction, which is a *biomechanical* phenomenon. Its surface interaction with blood and endothelial lining is the subject of bioengineering, especially if the biodegradability of a stent or closure device plays a role. In addition, it is paradoxical if a ‘natural’ solution for disease of blood vessels or heart valves based on tissue engineering cannot be implanted in the least invasive way possible. Consequently, there is always a situation of interdependence based on the mix of scientific disciplines shaping the technology, and the application needed to generate the safest and most effective result in patients.

### 2.1. Imaging Physics and Postprocessing

CT scanning, ultrasound-Doppler, and MRI are a combination of physics and electronic engineering, but creating a useful image is pure software power. Today, imaging is a discipline in itself, combining signal processing and ultra-fast computational techniques. Acquiring reliable morphological and functional information on the target area is indispensable when making a decision on whether or not a MICS intervention is feasible in a particular patient. This type of imaging, usually based on CT angiography and echocardiography, provides the information for access, advancing tools and devices, and a basic reference for combined use with fluoroscopy. Special software enhances the images for sizing, 3D images, and navigation. Last but not least, imaging must either verify the success of the intervention or reveal the cause of a complete or partial failure. The frames in Figure 2 and Figure 3 show 3D perioperative images obtained by transesophageal echocardiography. 

### 2.2. Materials and Surface Engineering

Sheaths for guiding tools and catheters should enable frictionless movement. Depending on the job in hand, varying degrees of bendability, steerability, or stiffness are required. Materials can respond to body temperature or be combined with a memory shape when expanded. Despite the abundance of alternative materials, the use of alloys such as Nitinol is most common for reasons of manufacturing and predictable behavior. The use of innovative materials requires laborious testing and follow-up in order to confirm their safety and effectiveness. Apart from surface treatments, the backbone of devices comprises conventional materials such as polymers, metal alloys such as stainless steel or titanium, and fabrics such as Dacron and polypropylene clones. Tissue engineering uses scaffolds that are biodegradable or natural material constructs, which enable and promote natural substitution for implants such as stents, clips, and valves. The use of this new class of materials is intended to improve biocompatibility and promote natural repair. This avoids the use of protective medication such as anticoagulants, and protects against postoperative and prosthetic infections.

### 2.3. Biomechanics

Gaining access to the heart for MICS comes with difficulties because cannulas are rather rigid, which complicates movement of the device. To overcome this difficulty, bendable cannulas are being developed to help fluent maneuvering of the device without reducing the safety of the surgery [7]. Another example is the soft tissue retractor, which maximizes access without injury to structures. Appendix A show examples of such innovations.Finite element analysis (FEA) and other numerical tools are gaining in popularity and importance in the process of medical device development in general [8,9,10]. Relatively simple FEA models are used to guide the design and shaping of tools. More complex and multi-scaled models can even be used to predict in vivo and longer-term behaviors such as radial pressure induced by cannulas or the strength of the stent posts of a prosthetic heart valve [11,12].Interactions between structures and fluids such as shear stresses in branched or curved vascular structures provide an extra layer of insight into the performance of devices and techniques. The long-term performance of tissue-engineered aortic valves is being explored by fluid–structure–interaction (FSI) models, while long-term thrombotic or hemolytic events caused by cardiovascular devices are now also being numerically determined [13,14,15].Estimation of fluid interactions around the heart valves obtained from in silico models of reconstructions of 4D Flow MRI or CT scan images (Figure 4). 

**Figure 4 jcm-10-03877-f004:**
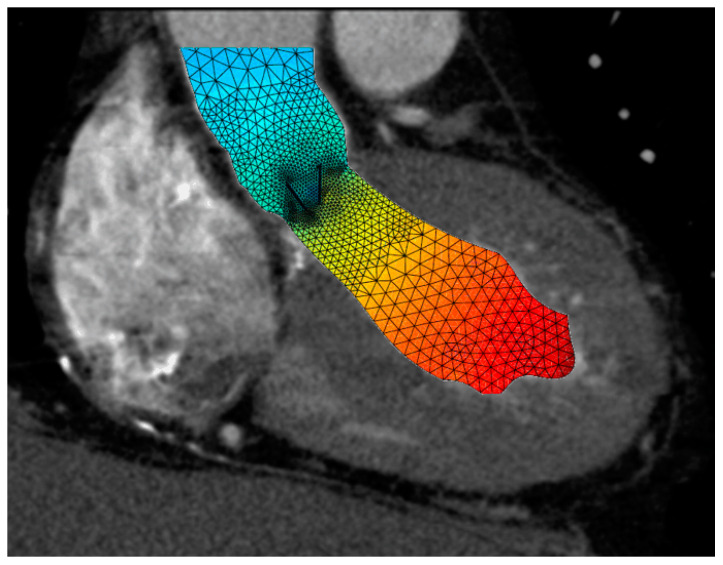
Sectional CT scan image of a hypertrophic left ventricle during systole, with a mechanical valve in situ showing the fluid velocity distribution, which is high (red) in the left ventricle and lower in the ascending aorta.

Detailed imaging and computational modeling can also predict whether a bypass or intraluminal repair of a coronary artery stenosis will be the best revascularization option (Figure 5) or to determine the effect of a certain anastomosis size pre-operatively (Figure 6).

### 2.4. Supporting Disciplines: Robotics, Machine Learning and Augmented/Virtual Reality

Engineering is primarily focused on delivering products and encompasses a wide variety of disciplines. Potential subdivisions of specialties may work for teaching and research purposes, but classification according to product group is the most common (e.g. aerospace, automation, and health care). What these groups have in common is that all efforts are directed at creating optimal efficacy and handling, which is covered by the term *usability*. In addition to usability, the term *human factor engineering* is used to describe how a product should be tailored to a particular type of user and the level of operation control that can be expected from that user. Therefore, complex technology is greatly supported by automation and microchips to assist the user. Complexity is inherent to an invention when high levels of reliability and safety are required. A simple example is braking and speed control in today’s cars, which are still under the control of the driver. However, in order to optimally distribute forces over the wheels while simultaneously applying real-time adjustments for road surface conditions, the process has become extremely complicated. Integrating self-driving cars into normal traffic is possible thanks to mapping and navigation. The technique is designed for the less-than-averagely skilled driver, and has resulted in an automated process and semi-automatic control based on multiple sensors, microchips, and a computer: the robot inside your car. From this perspective, it is possible to regard endovascular procedures based on detailed scans as a simple case for robotic control.

Aside from the mapping, navigation, and supervision of robotic assistance, the availability of machine learning enables real-time handling and adjustment of high-density imaging information. In other words, a robot can carry out jobs such as steering catheters and exact deployment of devices in accordance with preplanned scenarios. The surgeon then holds the contextual information and controls whether to proceed or abort the action. However, the surgeon must also assess imaging from a variety of sources such as echoscopy, CT scan, fluoroscopy, 3D adjustments, and real-time feedback in just a couple of seconds. The preparation or execution by means of a Microsoft HoloLense allows for the projection of the most relevant information, evenly matched with augmented or virtual reality components without searching, scrolling, or discussion [16,17]. Indeed, the work process, usability, appropriate skill level of the surgeon and human engineering factor determine whether the addition of these technologies indeed contributes to the efficacy and safety of a procedure and results in an additional benefit for the patient.

## 3. Advancement by Adaptation or Redesign: Unmet Patient Needs and Professional Demands

Innovations are viable only if well-described patient needs or professional demands are met. Although creating improvements looks simple and patient and professional interests should prevail, the complexity of technology and work process as well as timing of introduction are critical for acceptance. In the end, the expected benefits must be confirmed by clinical evidence.

### 3.1. Adaptation or Redesign

When the patient’s eligibility for a MICS procedure is established, the surgeon has to plan the procedure, which by definition is a multidisciplinary process. MICS requires adjusted procedures and adjusted tools, which are the result of focused design and manufacturing efforts. The MICS toolbox is especially designed and special-length tools can be matched with adapted suture techniques. Suturing can even be avoided by using sutureless devices [18]. The use of slave technology for motor-driven manipulation of the actuators is now widely accepted and is known as ‘robotic’ surgery. When accustomed to its use, the operator can benefit from stable vision, improved control of light source assistance, and specially designed end-tools, which allow for more degrees of freedom manipulation [19]. It can be concluded that MICS is an adaptation of conventional valvular and coronary surgery, based on engineering progress in the miniaturization of tools and sometimes devices, robotics, and imaging. In MICS, ECC and aortic cross-clamping are sometimes still necessary, despite ports and small incisions.

So-called micro-invasive cardiac surgery avoids the use of ECC and aortic cross-clamping and uses small or very small incisions for access to the heart. This includes the trans-apical approach, which enables aortic and mitral valve replacement, and procedures such as chorda repair of the mitral valve. These procedures are based on a radical redesign of the surgical process including implantation, redesign of tools, and redesign of devices.

In this respect, off-pump coronary artery bypass grafting (CABG) is still a micro-procedure in that ECC and aortic cross-clamping are avoided, but a sternotomy is still required. Addition of the use of small incisions or the use of ports and robotic slaves can make off-pump CABG more minimally-invasive, and even micro-invasive cardiac surgery, which is the aim of robot-assisted totally endoscopic bypass-grafting (TECAB).

### 3.2. Risk Reduction and User-Friendliness

MICS procedures tend to be time and resource consuming, which raises the question of cost-effectiveness. Here, again, biomedical engineering reasoning can be helpful. Whether it is aviation, the petrochemical industry, or information technology, the principle of acceptance of risks of failure at a level that is as-low-as-reasonably-achievable (ALARA) is a leading principle. By translating this principle into cardiac surgery, which can also be considered to be a high-volume, high-risk, high-reliability industry, any surgical procedure should be as least invasive as possible, considering that the risks are proven to be lower and the outcome is as good as that of conventional surgery. User-friendliness and reduction in complexity means a reduction in user errors and safer procedures. Examples of making surgical handling easier include adjusted optical 3D endoscopes, anastomotic connectors, sutureless valve prostheses, and also the silicone soft tissue retractor, which enables stable and wide access between the ribs without injury to muscles and neurovascular structures. However, user friendliness may be accompanied by less control in the short- or long-term. Visual control is reduced when implanting a sutureless valve, and anastomotic devices have been associated with reduced long-term patency.

The competition with percutaneous wire-supported technologies in the hands of cardiologists is a given fact. Although presented as carrying the lowest periprocedural risk and lowest costs including the supporting technology (short procedure, no general anesthesia, no postoperative ICU), the immediate and long-term effectiveness of percutaneous procedures may be less favorable when compared with surgery. Studies of five-year follow-up show that the initial advantage of PTCA and TAVR for certain indications is lost and the clinical outcomes for surgery and percutaneous procedures equalized [20,21]. 

### 3.3. Technical Reliability and Prediction of Outcome 

Biomedical engineering technologies are not only essential in reducing risk in performance and clinical use. Computational methods and repeat testing provide models of accuracy and reliability. Prior to designing a product, the acceptable fail and success range is established, and this has to be confirmed in bench tests and simulations. Therefore, strength, resistance to fatigue, and the behavior of heart valves in blood can be predicted, and the margins for failure are narrow. Specially-designed platforms often meet the specifications, but manufacturing tools and devices including the addition of microprocessors and software at the user end is extremely complex and expensive. Manufacturing reliability is not a specific biomedical engineering discipline, but is extremely important. It is not uncommon for 5–25% of products not to pass final quality control prior to packaging. Biomedical engineering techniques also promote the better prediction of success and failure, both in general and in individual cases. The computational models predict the performance and shear stresses of cannulas used in MICS. Based on angiography and CT angiography, the fractional flow reserve efficacy of a bypass can be calculated. These models include scenarios that are able to estimate the impact of either endoluminal repair or a bypass of the coronary artery stenosis in an individual patient. From both the clinical perspective and within the acceptable boundaries of available resources, patients can benefit from more ‘expensive’ procedures. The extra cost of technology must result in reliable predictions of immediate and sustainable improvement, must reduce risks to the minimum possible, and minimize interventional injury and patient discomfort. However, very few cost-effectiveness studies have been conducted in this field. This leaves us with a subjective observation that for a life-time event such as MICS, less discomfort, shorter hospital-stay, and earlier return to normal activities are worth the extra costs.

### 3.4. Therapeutic Alliance

Adaptations and inventions are driven by explicit customer feedback to manufacturing partners, either by invention or criticism. In other words, the current state of MICS and its further development depend on a ‘therapeutic alliance’ between profession, biomedical engineers, and industry in order to provide the technology for better treatments. However, the successful redesigning of the surgical process and the surgical toolbox by means of biomedical engineering concepts must be steered by surgeons. They know what is necessary and must conduct clinical research to prove the safety and effectiveness of the changes in equipment and adjustment of the work process. The role of industry—big or small—is primarily the provision of reliable manufacturing within the commercial boundaries of market needs and monitoring, not only of these boundaries, but also affordability and a return on investments. This menage-a-trois should be governed by the joint perception of a reasonable medical need and affordable patient benefit. The latter to be acceptable to both the medical profession and the public.

## 4. What to Choose from the Full Toolbox?

The novel application of smart materials and the availability of navigation and sensor technology and virtual reality and machine learning technologies exert continuous pressure to improve MICS. Considering the expense, only a few centers can afford to offer many of the possibilities offered by MICS, micro-invasive surgery, and transcutaneous solutions. 

### 4.1. What Is Best for Patients?

Although there are no exact figures, MICS for aortic and mitral valve interventions and their transcutaneous equivalents are commonly conducted in most cardiac surgical centers. The use of a robotic slave in aortic valve replacement and CABG is rare, but its use in mitral surgery is common [22,23,24]. Thanks to the availability of transcutaneous solutions, patients with an unfavorable benefit–risk profile for valve surgery can be treated. However, the many available options mean that linking a patient risk profile to an optimal benefit–risk profile has become more complex and time-consuming, and the need for clinical evidence is increasing. Assuming a solid hospital outcome registry is in place, and deep learning and artificial intelligence can assist in decision-making on the optimal procedure, given a certain degree of hospital experience.

However, from the engineering perspective, technical improvements in equipment and workflow should immediately translate into better skills, smarter surgical management, and risk reduction for patients. Apart from solving a problem, physician and patient experience determines the impact of an innovation in terms of clinical outcome and cost-effectiveness. For example, questions about the benefits of robotic assistance in MICS mitral surgery or the risks of transapical interventions without ECC are of a similar order of magnitude of debate to the percutaneous transluminal coronary angioplasty with or without stent (PTCA) versus CABG, off-pump CABG versus totally endoscopic coronary artery bypass (TECAB), and transcutaneous aortic valve replacement (TAVR) versus surgical AVR. Therefore, no quick answers based on clinical evidence are likely. Engineers can contribute to the debate by making products cheaper and more reliable. However, there is no substitute for the physician’s obligation to collect and share clinical experience, the paucity of which weighs heavier than ever, because clinical data finally determine the acceptability of choices in the best interests of the patient and reimbursement by insurers.

### 4.2. Training, Education and Learning Curve

Although the biomedical engineer can be held responsible for stocking the toolbox, using these tools is the privilege of competent and skilled operators. Engineers must describe and clarify the limitations of technology based on the description of design and testing. Specific usability studies must determine the minimum skill level required for safe and effective use. However, as both adapting and redesigning the technology is becoming increasingly complex, MICS technology requires special talents to work with video-assisted technology including responding to 3D and fused image information. This is in addition to managing the surrounding technology and alternative ways of tissue and tool handling. 

In other words, working with slaves, robots, imaging, and navigation assistance and handling the special MICS tools requires specialized training to master a learning curve [22,23,24]. Today, the manufacturer is required to provide adequate training and educational materials, but it is the surgeon’s responsibility to reach the proficiency level and to maintain experience either by performing a sufficient number of cases or by means of training in simulators. Although there is broad agreement on these training principles for claiming proficiency in MICS, team training on a simulator and a minimum case load are still not obligatory. All stakeholders must keep in mind that the progress of MICS often entails increasingly complex technology, which is doomed to fail without adequate education and training. 

### 4.3. Helping to Climb the Learning Curve

It is generally accepted that MICS for mitral valve surgery and TECAB has a rather steep learning curve [23,24,25]. The frontrunners in the field hint that proper selection and a time-consuming process of training are mandatory, but fail to describe how it should be conducted. However, an increasing number of journal reports are appearing that state that surgeons with experience can safely adapt to complex procedures, especially when robotic assistance is used. Task and work flow analysis are rarely used in surgery, but are common in industrial high-risk, high-performance, high-reliability professions. It shows that different levels of complexity demand different skills packages. Therefore, selection of surgeons and intervention cardiologists is critical in order to guarantee the safe and effective use of a technology. Simulators can select operators who demonstrate a good level of skills and capabilities in adopting new technologies [26]. However, the power of a good simulator to select a talented surgeon is rarely used. MICS beating heart simulators using animal or human hearts are available to train transapical mitral chordae repair, transcatheter valve implants including valve-in-valve, and novel anastomotic techniques for coronary arteries (See Figure 7) [27]. 

## 5. From Invention to Market Application

Today, MICS is a well-established procedure and is supported by effective tools and devices. Improvements in technology are often presented to colleagues by means of case studies and how-to-do-videos, which reflect the variation in local institutional experience in a certain area of application. Good examples of the challenges and what technology can do in dedicated hands are illustrated in the articles by Balkhy and Badhwar [22,23].

### 5.1. Robotic Assistance, Image-Guided Navigation, and Then?

It takes considerable resources and funding to convert a good idea into a tangible invention. An early analysis of the work process and the human factor can reveal if there is room for the innovation and, if so, what its impact will be. The following example aims to illustrate why an innovation often remains a small step ahead. During mini-coronary artery bypass (MI-CAB)/TECAB, the harvesting of the internal thoracic arteries (ITAs) is a cumbersome procedure, despite the use of a slave. In addition, suturing to create a vascular anastomosis between an ITA and a coronary artery on a beating heart is a challenge. Therefore, it is not surprising that engineers have been asked to assist in simplifying these tasks. A possibility that is currently being investigated is a small and smart vascular clip that connects the coronary artery and ITA graft, without the need of a shunt or the temporary occlusion of the coronary artery. The concept has already been investigated in neurosurgery for connecting a bypass to the median cerebral artery [28]. 

Special tools can make harvesting of the ITA more do-able, but it still requires extra time and skills. Imagine a bendable rail that is adjustable to the thoracic cavity and loaded with sensors to steer a ‘fully robotic cutting wagon’ with high-frequency energy under the ITA ‘roof’. There is no doubt that it could be developed, but at what cost, and more importantly, what are the extra benefits or savings that could justify the additional costs? The two answers are that a ‘robot cutting-rail’ must be a breakthrough in MI-CAB/TECAB, and that the scale of its use must have an impact on cost–benefit balance.

### 5.2. Competing Technologies and Decreasing Benefit

The next complicating factor in this field is competing technology of new stents and improved anastomosis technology. From the point of view of resources in R&D and return on investment, the smart clip and the ITA robot cutting rail are competitors. So the questions is, which challenge should be handled first. Assuming that the smart-clip technology is simpler and cheaper and can also be applied to vascular anastomoses in other areas, it is likely that we will have to wait for the robot cutting-rail. Therefore, surgeons still need to ask for better adaptive tools with slave support, instead of an autonomous robot for dissecting the ITA. The time saved by using the smart anastomotic clip will make up for the extra time taken for harvesting the ITA. This is called human factor engineering (i.e. finding the optimal interaction between human factor control and technology).

However, the substitution of a clip for suturing an anastomosis raises questions regarding sizing, flow capacity, flow patterns, and long-term patency. All these factors are influenced by the design, use of materials, and predictability of flow patterns by means of computational fluid dynamics (CFD) modeling (see Figure 5 and Figure 6). However, the technology can be reliably validated in experiments with isolated animal hearts and coupled with findings by angiography and CT angiography in animals and humans. This will result in revascularization scenarios for a patient-tailored clip, the flow performance of which can be predicted and would be considered safe and effective for bypassing the stenosis.

### 5.3. Other Limitations on the Pathway from Invention to Product

In the preceding sections, we mentioned factors such as market potential, work process, adaptation versus radical redesign, and human factor engineering. However, redesigning a work process in order to avoid complications of surgery by choosing a transcutaneous and wire approach also faces serious and costly challenges. It is foreseeable that based on the scale, competition, and state of the technology, prices of equipment or devices can and will decrease, similar to what happened with coronary stents and endovascular implant devices. However, in the beginning, the financial risks and the costs of R&D are high (Figure 8). The pressure for a short time-to-market from feasibility to commercial application is continuously present as it is the most significant cost-saver.

This pressure may affect the R&D team’s decisions when taking the hurdles of the numerous tests and the acquisition of clinical data as fast and efficiently as possible. However, sound and critical technical testing in the early development phase of a tool or device is the best risk management option as it avoids later and costly disappointments. Unbiased feedback by physicians who understand what is at stake is indispensable, and these users must be part of the development team. An example of the workflow of the process from idea to certified end-product can be found in Appendix A.

### 5.4. Technical Reliability versus Clinical Evidence

Clinical evaluation is the cornerstone of the confirmation of the safety and effectiveness of equipment and devices. Today, the organization of a simple open-label cohort study requires strict patient-protection warranties, and obtaining ethics approval for multi-center trials is time-consuming. The current European medical device regulation (MDR) requires credible clinical data both pre-and post-CE certification for devices and tools used in MICS [29]. Comparing treatment strategies such as MICS versus clip repair in mitral valves and also critical tools such as optics is time-consuming and complex. However, patients and physicians are entitled to knowledge of the level of safety and effectiveness of a procedure that is supported by the ‘newest’ technology. With the current knowledge, the reliability and sustainability of tools and devices for MICS can be accurately predicted, as can the behavior of implants. In other words, the boundary conditions for the safe and effective use of tools and procedures can be sufficiently tested. These tests must include safety of use in the hands of well-trained surgeons who have been carefully selected by working on near-vivo. The reliability of tools and devices can be sufficiently established based on the provenance of bench tests and technical and use performance in simulators. Hence, the transition to first-in-man experience becomes less risky for patients and avoids early clinical disappointment. Despite limited clinical experience, competent and trained operators can safely use novel tools and devices thanks to thorough technical testing and simulations with users. This results in rapid and cheaper availability for patients, as today, the collection of clinical data on a trial and error basis has become cumbersome. 

### 5.5. Future and Challenges for MICS

MICS is the result of longstanding improvement in cardiac surgical concepts and continuous and critical feedback by means of short- and long-term follow-up. The availability of wire-mounted devices has conquered a use-bias as first choice treatment option. Despite hard-to-beat low mortality and morbidity rates, resulting in excellent cost-effectiveness appreciation, these surgically look-alike devices are indeed implanted with very minimal access injury and without general anesthesia and intensive care admission at lower peri-procedural risks. The benefit of wire-mounted devices is gained easiest in the case of vascular access and a straightforward and permanent solution. Therefore, PTCA and TAVR are indeed for many indications as good or better than MICS solutions. However, in many cases, patients who undergo wire-treatments face an incomplete and temporary fix with doubts about its longstanding efficacy. Both classes of procedures are costly and therefore careful assessment of the patient’s risk profile is mandatory in order to advise the optimal procedure. An advantage of MICS is that multiple repair and replacement technologies can be used in a single procedure and the tools and devices are relatively cheap compared to wire-technology.

Other denominators for success are experience, surgical skills, and risk awareness, confirmed by long-term follow-up [30,31]. Conventional surgery for mitral valve disease still remains a procedure with very good results in experienced hands [32]. On the other hand, being aware of the risks associated with MICS is the key to risk control, which enables safe and good results in patients with pulmonary hypertension [33]. Although Ko et al. described excellent safety outcomes in a large cohort of patients, their article also pointed out rare risks that deserve better control such as bleeding [34]. Smart devices can facilitate surgery in many ways either by smoothening the implantation or skipping a process step when rapid deployment devices are used for valve replacement, similar to wire-mounted devices [35]. 

In summary, the effectiveness and acceptance of MICS can still improve by means of patient tailored mapping of the risk profile and predicting the effect of the procedure. A MICS procedure is usually more complete and durable than a wire intervention. Registries and artificial intelligence will certainly become of great support to make this MICS case for complete and durable solutions at a low and acceptable risk. Imaging, simulations, and navigation are essential for per-operative risk control. Simulations based on conventional imaging can help to choose the optimal valve-replacement device and/or prevent injuries to the left ventricular outflow tract including obstruction by anterior leaflet motion by the mitral valve. Smart and minimal slaves will become affordable additions to improve surgical handling. However, the standards for designing this technology with high levels of reliability and user friendliness have become hard to meet, apart from the challenges to be managed before an idea becomes a device, which can be made commercially available. Appendix A illustrates this complex and laborious process, which bears a common message that only well tested and critically challenged technology will work. From the clinician’s perspective, well known hurdles preventing the collection of clinical experience are privacy rules, Ethic Review Boards, and transparent funding structures. Developers, biomedical engineers, and clinicians face the challenge to jointly determine when a new technology is sufficiently usable and effective in order to simplify complexity and to improve the MICS workflow and safety. 

Awareness of the existence of a steep learning curve and need of specific surgical competence for MICS are key conditions. Although the learning curve per se is not very well established, Holzhey et al. [25] described the challenges and recent publications have emphasized the selection of capable surgeons and dedicated training embedded in supervised exposure by experienced peers [36]. In other words, biomedical engineering and technology help, but cannot compensate for deficits in surgical competence and training.

## 6. Conclusions

Surgeons should be aware of the biomedical engineering principles conditioning their MICS technology. Innovative technology for MICS can add to the complexity and create challenges for optimal control by means of imaging or automation. Awareness of human factor elements, work process, and systems control should determine how a technology or tool is implemented in MICS in clinical practice. Surgeons should assist biomedical engineering in many phases of R&D and engineers must assist surgeons for the purposes of surgical training. This collaboration enables thorough preclinical testing in established biomedical engineering models and should result in technical reliability and safe use in appropriately selected and trained hands. Under these conditions, good engineering can be a substitute for limited clinical experience.

## Figures and Tables

**Figure 1 jcm-10-03877-f001:**
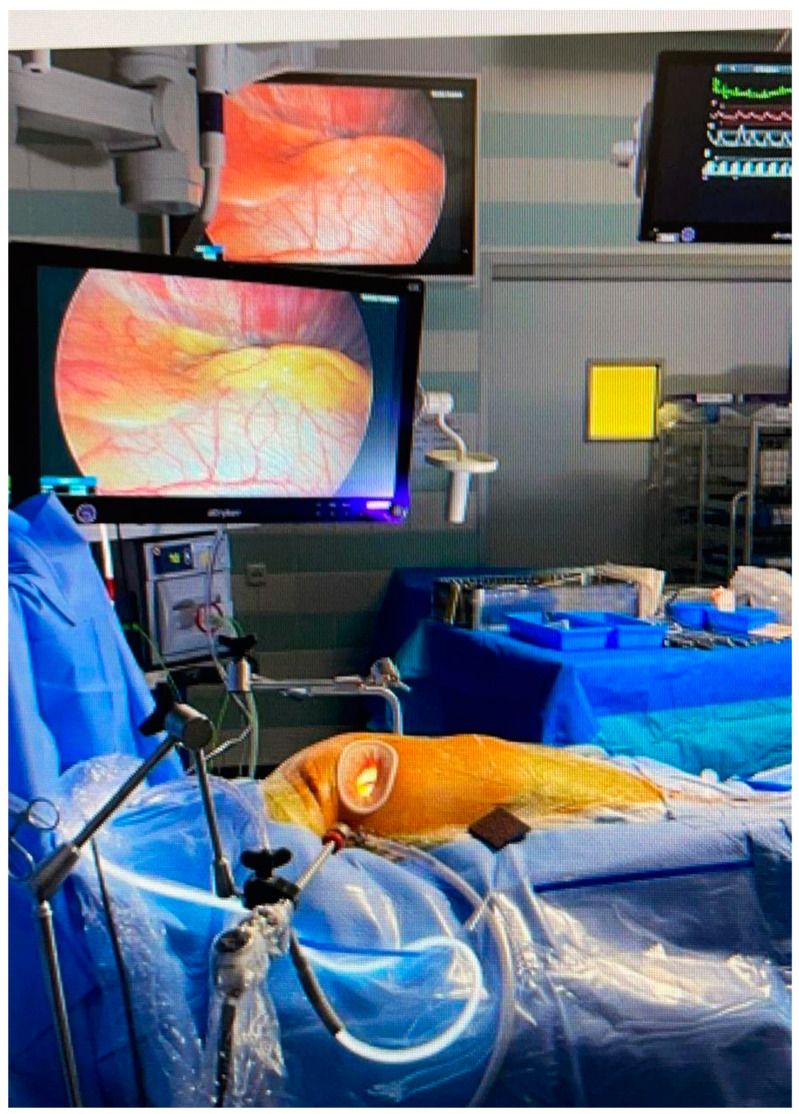
Overview of MICS setup. Right thoracotomy approach with camera and light on intact pericardium. Working port with soft tissue retractor providing access for instrumentation.

**Figure 2 jcm-10-03877-f002:**
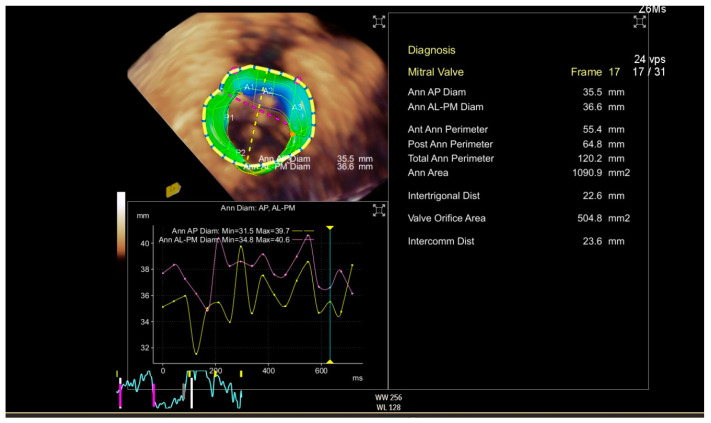
Preoperative 3D transthoracic echocardiography (TTE) displaying the anatomical information with a focus on annular diameters.

**Figure 3 jcm-10-03877-f003:**
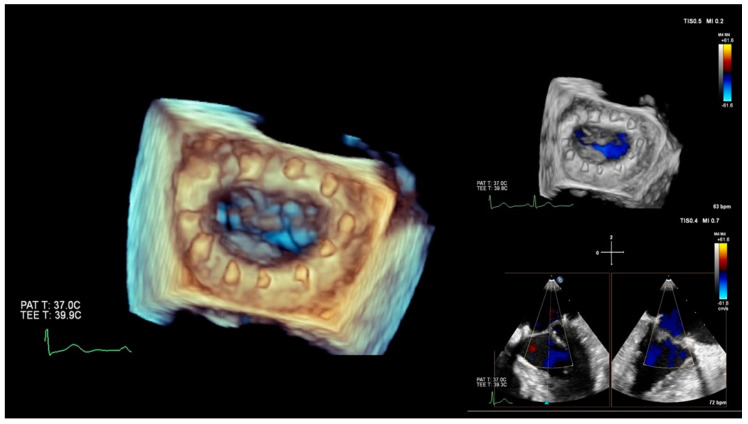
Preoperative 3D transesophageal echocardiography (TEE) and Doppler after implantation of an annular ring. Note suture knots and absence of regurgitation (blue) in various cross-sections.

**Figure 5 jcm-10-03877-f005:**
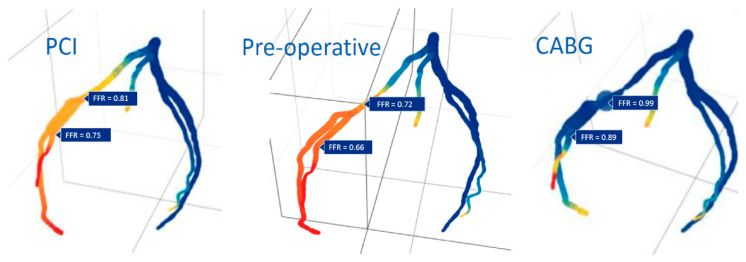
Simulation of FFR changes and flow adjustment after PCI and CABG compared to the pre-operative baseline. CABG yielded better FFR values and a better profile of this simulated condition (blue), although peripheral low-flow areas remained (red and yellow).

**Figure 6 jcm-10-03877-f006:**
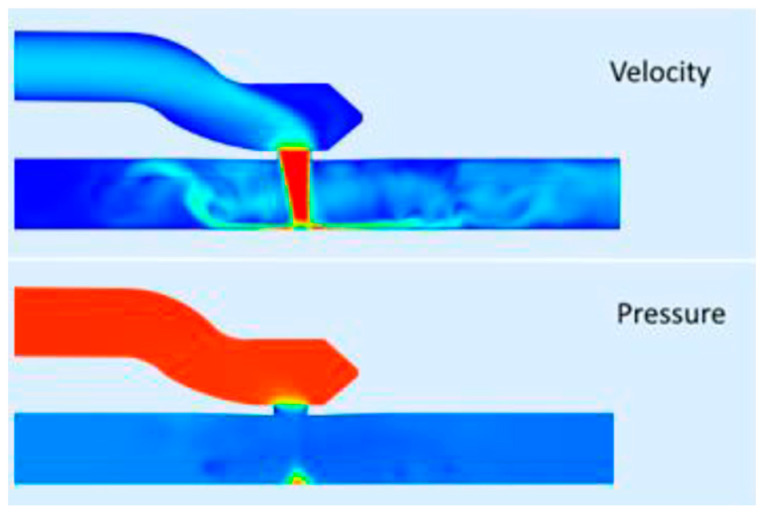
Example of a computational fluid dynamics (CFD) simulation to determine blood velocity and pressure drop in a LIMA-LAD bypass anastomosis. Top image: Red: high velocity, blue: low velocity. Bottom image: Red: high pressure, blue: low pressure.Courtesy of AMT Medical BV Netherlands.

**Figure 7 jcm-10-03877-f007:**
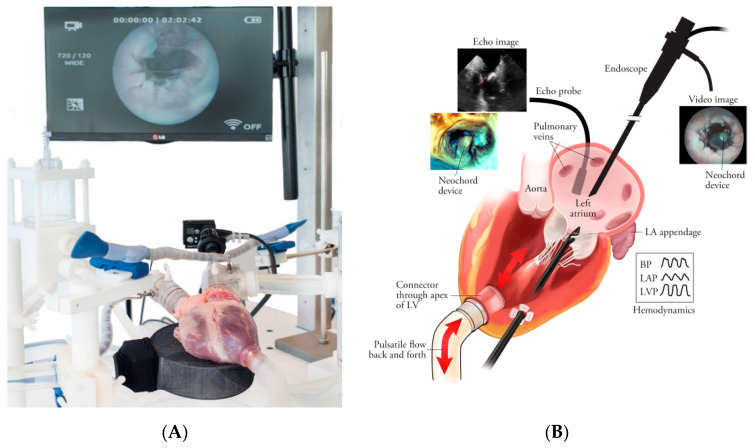
(**A**) Cardiac BioSimulator provides a surgical training platform with realistic haptic tissue feedback, real-time hemodynamic data visualization, and visual feedback such as endoscopic videos, (Doppler) ultrasound, fluoroscopic images, and high-speed camera images. (**B**) Schematic overview of the Cardiac BioSimulator. LifeTec Group BV, The Netherlands. (**C**) Surgical training being performed on the Cardiac Biosimulator. LifeTec Group BV, The Netherlands.

**Figure 8 jcm-10-03877-f008:**
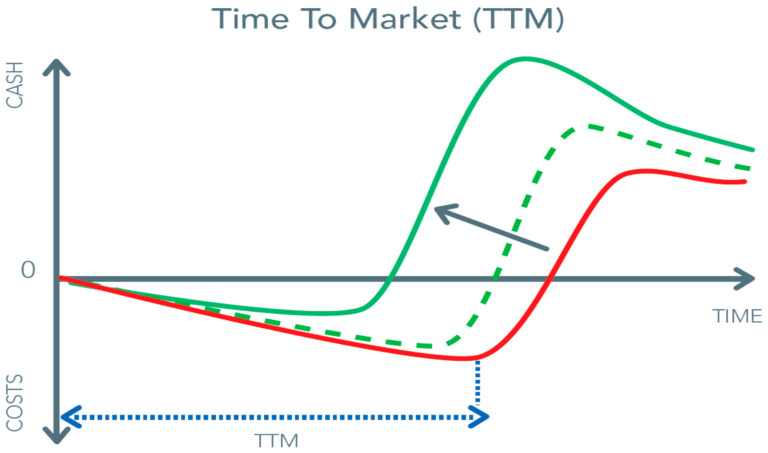
Reduction in time to market (TTM) moves the red curve to the left. The red curve presents the TTM situation without proper usage of biomedical engineering.The green curve presents a simultaneous reduction in duration and costs of investment, while the green dashed line indicates how the TTM process will change when moving from the red line to the green solid line.

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
