# Peer review of "The Impact of Biomedical Engineering on the Development of Minimally Invasive Cardio-Thoracic Surgery"

_jcm, 2021, doi:10.3390/jcm10173877_

Round 1

Reviewer 1 Report

The present overview is a very well written paper about the future of minimally invasive cardiac surgery.

The Authors shoudl be commended for their effort because they did a great summary with many great figures and jumps on the future

I have only one suggestion:

The review lacks of references on current experiences of minimally invasive surgery. In particular, data regarding the difficulties and learning curve of new minimally invasive program. 

Please, improve the manuscript with a section dedicated to available experince (i.e. PMID: 29707310  or PMID: 30179623) and how these experiences can be improved in the next future

Author Response

Dear Editorial Team,

We acknowledge the reviewers’ comments for improvement of the manuscript.

We added to-the-point considerations and recent publications on MICS in section 5.5 (see below). In this section we address the challenges to maintain safety and effectiveness of MICS compared to wire-interventions.

We recognize the learning curve element, which we addressed by pointing at the possibility to acquire experience in simulators. We concluded section 5.5 with a strong statement and references associated with climbing the learning curve.

Last but not least we add a process -flow chart for device / technology development by means of Figure 11, which we included in the supplement with other figures as was suggested by the reviewer.

Many thanks for taking care of our manuscript on behalf of all authors,

Bertus.

Revised section 5.5:

5.5. Future and challenges for MICS

MICS is the result of longstanding improvement of cardiac surgical concepts and continuous and critical feedback by means of  short- and long-term follow-up. The availability of  wire-mounted devices has conquered a use-bias as first choice treatment option. Despite hard-to-beat low mortality and morbidity rates, resulting in excellent cost-effectiveness appreciation, these surgically look-alike devices are indeed implanted with very minimal access injury and without general anesthesia and intensive care admission at lower peri-procedural risks. The benefit of wire-mounted devices is easiest gained in case of vascular access and a straightforward and permanent solution. Therefore PTCA and TAVR are indeed for many indications as good or better than MICS solutions. However, in many cases patients who undergo wire-treatments face an incomplete and temporary fix with doubts about its longstanding efficacy. Both classes of procedures are costly and therefore careful assessment of the patient’s risk profile is mandatory in order to advice the optimal procedure. An advantage of MICS is that multiple repair and replacement technologies can be used in a single procedure and the tools and devices are relatively cheap compared to wire-technology.

Other denominators for success are experience, surgical skills and risk awareness, confirmed by long term follow-up [30,31]. Conventional surgery for mitral valve disease still remains a procedure with very good results in experienced hands [32].  From the other hand being aware of the risks associated with MICS is the key to risk control, which enables safe and good results in patients with pulmonary hypertension [33]. Although Ko et al. describe excellent safety outcomes in a large cohort of patients, their article also point at rare risks which deserve better control such as bleeding  [34]. Smart devices can facilitate surgery in many ways either by smoothening the implantation or skipping a process step when rapid deployment devices are used for valve replacement, rather similar to wire-mounted devices [35].

Summarizing, the effectiveness and acceptance of MICS can still improve by means of patient tailored mapping of the risk profile and predicting the effect of the procedure. A MICS procedure can usually more complete and durable than a wire intervention. Registries and Artificial Intelligence  will certainly become of great support to make this MICS - case for complete and durable solutions at a low and acceptable risk. Imaging, simulations and navigation are essential for per-operative risk control. Simulations based on conventional imaging can help to choose the optimal heart-replacement device and/or prevent injuries to the left ventricular outflow tract including obstruction by anterior leaflet by anterior leaflet motion by the mitral valve. Smart and minimal slaves will become affordable additions to improve surgical handling. However, the standards  for designing this technology with high levels of reliability and user friendliness have become hard to meet. Apart from the challenges to be managed before an idea becomes a device, which can be made commercially available. Figure 11 (supplement) illustrates this complex and laborious process, which bears  the common message that only well tested and critically challenged technology will work. From the clinician’s perspective well known hurdles preventing the collection of clinical experience are privacy rules, Ethic Review Boards and a transparent funding structures. Developers, biomedical engineers and clinicians face the challenge to jointly determine when a new technology is sufficiently usable and effective in order to simplify complexity and to improve the MICS workflow and safety.

Awareness of the existence of a steep learning curve and need of specific surgical competence for MICS are key conditions. Although the learning curve per se is not very well established Holzhey et. al. [25] described the challenges and recent publications  emphasize selection of capable surgeons and dedicated training embedded in a supervised exposure by experienced peers [36]. In other words, biomedical engineering and technology help but cannot compensate for deficits in surgical competence and training.

Reviewer 2 Report

This is a nicely written review by Cocchieri et al describing the boundary conditions for minimally invasive cardiac surgery (MICS) and the aim to reduce procedure-related patient injury and discomfort. They also discuss the MICS work process and the demand for improved tools. This review also cover innovations which can represent a desired adaptation of an existing work process or a radical redesign of 18 procedure and devices. The authors also discuss over the safety, feasibility, efficacy and cost-benefit of implementing new technologies in clinical practice.

The first part of the manuscript is mainly a narrative review describing the history of technological achievements in cardiac surgery. The authors then provide specific examples of these innovations and they conclude with interesting perspectives. There are certain points that could be taken into consideration and possibly increase the interest of the reader in this review.

The authors provide some nice figures of these new innovations; however, I feel it would be more useful if they could design one (or more) figure depicting the new innovation and the old technology or approach that is being replaced or amended. This would give a global view of all these technologies and how these are being applied into practice. Most of the current figures could be moved to supplemental.

Figure 10 is nice but could be possibly augmented if another figure showing the workflow along with the timeline.

Finally, it would be intriguing if the authors could share some future perspectives and directions based on experimental/preclinical studies.

Author Response

(The authors gave the same response as above.)
